# m^6^A mRNA Destiny: Chained to the rhYTHm by the YTH-Containing Proteins

**DOI:** 10.3390/genes10010049

**Published:** 2019-01-15

**Authors:** Ditipriya Hazra, Clément Chapat, Marc Graille

**Affiliations:** Laboratoire de Biochimie, Ecole polytechnique, CNRS, Université Paris-Saclay, 91128 Palaiseau CEDEX, France; ditipriya.hazra@polytechnique.edu (D.H.); clement.chapat@polytechnique.edu (C.C.)

**Keywords:** epitranscriptomics, mRNA methylation, m^6^A readers, YTH domain

## Abstract

The control of gene expression is a multi-layered process occurring at the level of DNA, RNA, and proteins. With the emergence of highly sensitive techniques, new aspects of RNA regulation have been uncovered leading to the emerging field of epitranscriptomics dealing with RNA modifications. Among those post-transcriptional modifications, N6-methyladenosine (m^6^A) is the most prevalent in messenger RNAs (mRNAs). This mark can either prevent or stimulate the formation of RNA-protein complexes, thereby influencing mRNA-related mechanisms and cellular processes. This review focuses on proteins containing a YTH domain (for YT521-B Homology), a small building block, that selectively detects the m^6^A nucleotide embedded within a consensus motif. Thereby, it contributes to the recruitment of various effectors involved in the control of mRNA fates through adjacent regions present in the different YTH-containing proteins.

## 1. Introduction

The regulation of gene expression plays a central role during development and upon cell response to stress exposure. Hence, living organisms have developed highly complex mechanisms at different steps of gene expression to tune various cellular pathways. Many of those events occur at the post-transcriptional level through the formation of protein-RNA complexes that will influence various aspects of messenger RNA (mRNA) maturation such as alternative splicing, editing, export, and polyadenylation. Therefore, RNA binding domains (RBDs) are key actors in these regulatory mechanisms through their recognition of specific RNA sequences or structures. The most common RBDs are the RNA recognition motif (RRM) (corresponding to about 2% of all RNA binding domains), the hnRNP K Homology domains (KH), Piwi Argonaute and Zwille domains (PAZ), and double stranded RNA-binding domains (dsRBD) [1,2,3,4]. With recent technological developments combining UV cross-linking together with oligo(dT) purification of mRNAs, we now have comprehensive lists of RNA-binding proteins in various eukaryotic organisms [5,6,7,8], indicating that many other RBDs are yet to be uncovered.

The recent identification of several internal post-transcriptional modifications such as m^6^A (N6-methyladenosine), m^1^A (N1-methyladenosine), pseudouridine (Ψ), m^5^C (5-methylcytosine), and ac^4^C (N4-acetylcytosine) within mRNAs has shed light onto an additional layer of regulation now known as epitranscriptomics [9,10,11,12,13,14,15,16,17,18]. Indeed, similar to the dynamic modifications known to occur in DNA and proteins, epitranscriptomics modifications widely contribute to the regulation of biological pathways [19,20]. At present, the most studied mRNA modification is m^6^A. This modification was initially identified in mRNAs four decades ago [9]. However, because m^6^A does not alter base pairing [21], and hence does not result in the introduction of a stall or a mutation during reverse-transcription, the development of advanced techniques to precisely map m^6^A sites has been a main obstacle for studying its biological significance. This field was reignited in 2011, following the discovery of the m^6^A demethylase FTO (fat mass and obesity-associated protein), a protein involved in human obesity, as a so-called ’eraser‘ enzyme that removes m^6^As present on mRNAs [22]. More recently, several laboratories have finally been successful in mapping m^6^A at individual-nucleotide resolution using cross-linking and immuno-precipitation with m^6^A-specific antibodies. High-throughput sequencing of the immuno-precipitated RNA fragments revealed the presence of more than 10,000 m^6^A sites in human cells, affecting more than 25% of the transcriptome. Their detailed mapping showed an enrichment of m^6^A near the stop codon and in the 3′ untranslated region (3′ UTR) of the target mRNAs. The main ‘writer’ methyltransferase is a multi-protein complex composed of at least METTL3, METTL14, WTAP, and KIAA1429, which is responsible for m^6^A deposition on the consensus motif DRA*CH (where D is A, G or U; R is A, or G; A* is the methylated A and H is A, C ,or U; [10,13,23,24]). The m^6^A marks can be deleted by ‘erasers’ such as FTO and ALKBH5 [22,25]. Furthermore, this modification can attract m^6^A-binding proteins known as ‘readers’, as well as repel various proteins regulating mRNA functions [10,26,27,28]. This differential recruitment of regulatory proteins on m^6^A marks subsequently determines the fate of m^6^A-containing mRNAs, such as splicing, translation, degradation, or cellular localization. The most studied RNA binding module known to directly recognize m^6^A marks is the YTH domain. Indeed, many studies are scrutinizing eukaryotic YTH-containing proteins in order to clarify their roles in the regulation of mRNA fates. The present review aims at summarizing our current knowledge on this class of m^6^A readers.

## 2. The YTH Domain, an m^6^A RNA Grip

The foremost member of this m^6^A reader protein family is human YT521-B (hereafter termed YTHDC1), which was initially identified as a factor interacting with Tra2β, SC35, SF2, hnRNP G, and SAM68 splicing factors [29,30]. Despite this clear interaction of YTHDC1 with splicing machinery, no known RNA binding domain was detected and the only common feature with splicing factors was the presence of repeats of charged amino acids [30]. Further bioinformatics analyses led to the identification of an additional conserved region, named YTH (for YT521-B Homology) domain. This domain was found exclusively in eukaryotic proteins from fungi (one member in *Saccharomyces cerevisiae* and *Schizosaccharomyces pombe* yeasts) through plants (13 members in *Arabidopsis thaliana*) to higher eukaryotes (five members in human, namely YTHDC 1–2 and YTHDF 1–3) [31,32]. Although this domain was predicted to be a putative RNA binding domain since its identification, its RNA binding property was only demonstrated in 2010 when Zhang et al. showed that YTHDC1 can bind degenerate RNA sequences [33]. YTH-containing proteins became a topic of strong interest when YTHDF1, 2, and 3, three paralogous members of the YTH-containing protein family, were found to be the most enriched human proteins specifically retained by an RNA fragment containing an m^6^A modification at the heart of the DRA*CH consensus sequence [10,26].

The YTH domain is made of 150 to 200 residues and adopts an α/β fold, with four to five α-helices surrounding a curved six stranded β-sheet as revealed by the NMR structure of YTHDC1 YTH domain determined by the RIKEN Structural Genomics and Proteomics Initiative in 2007 (PDB code: 2YUD). At the center of the β-sheet lies a cavity delineated by conserved hydrophobic residues including three tryptophan amino acid side chains (in some YTH domains, one Trp residue is substituted by either Leu or Tyr) forming a so-called aromatic cage (Figure 1A). Several 3D-structures of different YTH-m^6^A containing RNA complexes have revealed that this aromatic cage is responsible for the specific recognition of the m^6^A mark by YTH domains [34,35,36,37,38]. Indeed, in the case of human YTHDC1 bound to an m^6^A containing RNA, the aromatic cage specifically accommodates m^6^A via the side chains from W377, W428, and L439 amino acids [37]. The m^6^A methyl group forms methyl-π interaction with W428 side chain while the purine base is sandwiched between the W377 and L439 side chains (Figure 1A). This interaction of the m^6^A methyl group with W428 most probably explains the higher affinity (20 to 50-fold difference) of YTH domains for an m^6^A containing RNA oligonucleotide compared to the same unmodified oligonucleotide [36]. Sequence alignment revealed that the W377 and W428 residues are strictly conserved among the YTH-containing proteins while the position corresponding to L439 can be occupied by either Leu, Tyr, or Trp. The integrity of this aromatic cage is essential for the recognition of m^6^A since mutation of any of these three residues to alanine disrupts binding [34,37,38]. Polar amino acids (S378, N363, and N367 in human YTHDC1) also contribute to m^6^A recognition by forming specific hydrogen bonds with nitrogen atoms from the adenine base and then participating together with the aromatic cage to the selective recognition of m^6^A (Figure 1A). Residues surrounding this aromatic cage are also important for RNA binding as they form a large positively-charged surface interacting with RNA phosphate groups from nucleotides surrounding m^6^A and also provide some specific interactions with bases. A large majority of m^6^A marks generated by the main m^6^A methyltransferase ‘writer’ is found within Gm^6^AC (70%) or Am^6^AC (30%) motifs [37]. Interestingly, structural analyses coupled to isothermal titration calorimetry (ITC) measurements using various RNA sequences have shown that the affinity of YTH domain from YTHDC1 for RNA fragments is higher (five- to six-fold difference) when the nucleotide immediately upstream of m^6^A is a G compared to an A [37,38], suggesting the co-evolution of both the active site of METTL3-METTL14 methyltransferase holoenzyme and of the RNA binding site of human YTHDC1. Indeed, in the structure of YTHDC1 bound to a GGm^6^ACU RNA fragment [37], the carbonyl group at position six of the G base preceding m^6^A forms a hydrogen bond with the main chain nitrogen group from V382 (or equivalent position in other YTH domains; Figure 1A). The amine group, present at the same position in A, is less prone to form such hydrogen bond, which could rationalize the differences in measured affinities. However, this seems to be specific of YTHDC1 as all other tested YTH-containing proteins (YTHDF1/2, YTHDC2, and *S. cerevisiae* Pho92) exhibit similar affinities for RNAs containing any of the four nucleotides at the position immediately upstream m^6^A [38]. This probably results from the fact that this nucleotide binds into different pockets with no obvious base specificity on YTHDF1 or the *Zygosaccharomyces rouxii* Mrb1/Pho92 fungal orthologue compared to YTHDC1 [35,38]. So far, structural studies have been performed only with 5- or 7-mers RNAs, which are known to bind more weakly (Kd values in the 2 to 30 µM range) than longer RNAs (9 or 16-mers for instance; Kd values ranging from 0.2 to 0.3 µM for YTHDC1 up to 7.5 µM for YTHDC2 [38]). Therefore, additional structural studies with longer RNA fragments might be of interest to bring more information on RNA recognition by these YTH motifs.

In summary, YTH-domain containing proteins interact with single-stranded RNAs and selectively identify the presence of a modified m^6^A nucleotide at the center of a consensus signature motif matching that of the major m^6^A methyltransferase machinery identified to date.

## 3. YTH, a Building Block Governing the Fates of m^6^A Containing mRNAs

Most RNA binding modules are embedded within larger proteins and are surrounded by either structured domains or low complexity regions, all with various functions [8,40,41]. The YTH domain is no exception to this rule and clearly serves as a building block mostly flanked by regions predicted to be disordered (Figure 2; [42,43,44]). By influencing both the sub-cellular localization of these YTH proteins and their partners, these flanking regions are thereby important for the function of YTH proteins in the regulation of m^6^A-containing RNA fates, i.e., splicing, mRNA nucleocytoplasmic export, translation and mRNA decay [26,35,45,46,47].

As mentioned above, bioinformatics analyses have identified 13 YTH-containing proteins in *A. thaliana*, which roles are being clarified. As this topic has been nicely discussed by Bhat et al. in this special issue [32], these plant YTH proteins will not be discussed in this section. On the basis of amino acid sequences, YTH domain proteins from different vertebrates can be divided into three families: YTHDC1, YTHDC2, and YTHDF1–3. There is no significant sequence similarity between members from those different families outside of the YTH domain, hence YTHDC1, YTHDC2, and YTHDF1–3 members cannot be considered as paralogues.

### 3.1. YTHDC1

YTHDC1 (previously known as YT521B) is the founding member of YTH family of proteins. In this protein, the YTH domain is the only region predicted to be folded and is surrounded by regions rich in charged residues (Glu-rich, Arg-rich and Arg-Asp-Glu-rich, or RED segments; Figure 2) or in proline (P-rich). The human protein contains nuclear localization elements and is found to localize in distinct subnuclear bodies, so-called YT bodies, adjacent to nuclear splicing speckles [48]. This nuclear localization is in agreement with its initially described interactions with various splicing factors (i.e., Tra2β, SC35, SF2, hnRNPG, and SAM68 [29,30]). Recent studies have clearly established a role of YTHDC1 in splicing both in human cells and in *Drosophila melanogaster* [45,49]. Indeed, this protein contributes to alternative splicing by binding to the pre-mRNAs and by influencing the splice site selection [33]. Mechanistically, YTHDC1 directly interacts with SRSF3 and SRSF10, two serine/arginine-rich splicing factors, in a competitive manner (Figure 3). In doing so, it enhances the binding of SRSF3 to targeted pre-mRNAs resulting in exon inclusion while precluding the binding of SRSF10, which is involved in exon skipping. This YTHDC1-mediated recruitment of SRSF3 is clearly dependent on m^6^A as either METTL3 silencing or a YTHDC1 double mutant (two Trp residues from the aromatic cage of the YTH domain are substituted by Ala) strongly reduce binding of SRSF3 to RNAs *in cellulo* [45]. As YTHDC1 has a 30-fold higher affinity for SRSF3 over SRSF10, it interacts predominantly with SRSF3 and hence may favor alternative splicing of m^6^A-containing pre-mRNAs. This interaction between YTHDC1 and SRSF3 is not only involved in alternative splicing but also in the polyadenylation process of the pre-mRNAs through their association with the pre-mRNA 3′ end processing factors CPSF6 [50]. Moreover, YTHDF1 and SRSF3 collaborate with NXF1 to drive efficient export of transcripts subject to m^6^A control from the nucleus to the cytoplasm (Figure 3; [47]).

YTHDC1 has also been shown to participate in the regulation of the abundance of MAT2A mRNA, which encodes a subunit of one of the methionine adenosyltransferases responsible for the synthesis of S-adenosyl-L-methionine (SAM) cofactor from methionine and ATP [51]. The MAT2A mRNA is one of the well-characterized targets of the recently identified METTL16 m^6^A RNA methyltransferase, which introduces m^6^A marks in the 3′ UTR of this mRNA [52], a modification crucial for mouse embryonic development [53]. Depletion of SAM was shown to enhance the removal of a retained intron in MAT2A pre-mRNA leading to induced expression of this mRNA [52] and to reduce m^6^A levels in MAT2A 3′ UTR [51]. Whether the role of YTHDC1 in this pathway is to favor splicing, nucleocytoplasmic transfer, or recruitment of mRNA decay enzymes remains to be clarified.

In fruit-fly, the YTHDC1 orthologue is also involved together with m^6^A methylation in the alternative splicing of Sex lethal (*Sxl*), which encodes a master regulator of sex determination and dosage compensation [49]. Although *Sxl* is expressed in males and females, the presence of an additional internal exon in males introduces a premature stop codon that results in both the production of a truncated and non-functional *Sxl* protein and the rapid elimination of the transcript most probably by the nonsense-mediated mRNA decay pathway [54,55]. Inactivation of either the METTL3 subunit of the m^6^A mRNA methyltransferase holoenzyme or the YTHDC1 protein in the female but not the male fly results in retention of the male-specific exon concomitant with the decrease of the female-specific isoform, clearly indicating a female-specific splicing defect linked to altered m^6^A deposition and recognition.

Beyond its role in splicing, human YTHDC1 also helps in transcriptional repression of X chromosome genes by X-inactive specific transcript (XIST), a long non-coding RNA that plays a critical role in inactivation of one X chromosome in female cells [56]. XIST is a heavily methylated RNA with at least 78 m^6^A sites and the preferential binding of YTHDC1 to m^6^A marks is necessary for XIST-mediated transcriptional silencing. The depletion of m^6^A ‘writer’ causes inhibition of XIST function and this defect can be restored by artificially tethering YTHDC1 to XIST in cells lacking m^6^A methylation machinery. A comprehensive understanding of the mode of action of YTHDC1 on XIST will necessitate further studies.

### 3.2. YTHDC2

Compared to the other YTH domain-containing proteins, where the YTH domain is embedded within low complexity regions, members of the YTHDC2 family are multi-domain proteins (Figure 2). Apart from the C-terminal YTH domain, there is a N-terminal R3H (arginine and histidine-rich) domain with RNA-binding property [57] preceded by a Gly-rich patch, a central DEAH-box helicase domain (where an ankyrin repeat domain is inserted in the middle of the second RecA domain), an helicase associated 2 domain (HA2), an OB-fold (oligonucleotide / oligosaccharide-binding fold) and a C-terminal extension (CTE) also found in human DHX36, a DNA/RNA DEAH-box helicase involved in G-quadruplex unwinding (Figure 2; [58]). In agreement with its domain composition, human YTHDC2 has RNA dependent ATPase and 3′→5′ RNA helicase activities [59,60,61]. This protein is mainly a diffuse cytoplasmic protein, but it is also enriched in peri-nuclear regions [57]. As expected from the presence of several RNA binding domains, YTHDC2 interacts with mRNAs and in particular with m^6^A-rich mRNAs through its YTH domain [60,62].

Initially, YTHDC2 was shown to associate with hepatitis C virus protein NS5B to facilitate viral DNA replication [59] and to play an important role in the proliferation of cancer cells by enhancing the translation of metastasis-related genes [63,64]. More recent studies have converged towards an important role of YTHDC2 in the progression of meiotic prophase I, which is a critical and long meiosis stage characterized by many chromosomal events that will ultimately lead to severing of the genome into two halves [60,65]. Consequently, the inactivation of YTHDC2 gene in mice results in gametogenesis defects and infertility [60,61,62,66].

In human cells, YTHDC2 interacts in an RNA-independent manner with the meiosis-specific MEIOC protein as well as with the 5′→3′ exonuclease XRN1 [57,60,62,65,67]. Whether YTHDC2 can interact simultaneously with both MEIOC and XRN1 is unclear but it is tempting to speculate that the MEIOC–YTHDC2 complex interacts with m^6^A-enriched mRNAs to address them to degradation by the XRN1 exonuclease. However, this model could be restricted to a subclass of mRNAs as various studies have observed different effects on mRNA translation and stability upon inactivation of YTHDC2 gene. Indeed, YTHDC2–MEIOC complex could stabilize meiosis-specific transcripts [67] while destabilizing mitotic mRNAs [60,62,65].

Conversely, YTHDC2 has also been shown to enhance translation efficiency of mRNAs concomitantly to decrease their stability (Figure 3; [62]). This could result from the association of YTHDC2 with the head of the 40S ribosomal subunit both at the level of the 40S but also of the 80S [57]. More precisely, YTHDC2 binding site on the 40S subunit maps in the vicinity of the mRNA entry and exit sites, which could rationalize the dual role of YTHDC2 in enhancing translation efficiency by recruiting m^6^A-containing mRNAs to the ribosome but also decreasing mRNA stability by recruiting XRN1 to those mRNAs (Figure 3).

### 3.3. YTHDF Family

Human YTHDF1–3 are cytoplasmic proteins made of a single C-terminal YTH-domain that binds to m^6^A marks separated from a N-terminal low-complexity domain by a segment rich in Pro, Gln, and Asn amino acids (Figure 2; [26,34,38]). Those three proteins are highly homologous with 65 to 68% and about 85% of sequence identity and similarity, respectively.

YTHDF2 was the first protein to be functionally characterized, especially regarding the repertoire of mRNAs that it binds to, as well as its mode of action. Photo-activable ribonucleoside crosslinking and immunoprecipitation (PAR-CLIP) combined with RNA-immunoprecipitation coupled to sequencing (RIP-Seq) experiments showed that human YTHDF2 selectively targets more than 3000 different transcripts and binds predominantly to their 3′ UTR and around the stop codon [26]. Furthermore, YTHDF2 knock-down results in the accumulation of its targets in translatable or actively translating polysome pools, pointing to a crucial role of YTHDF2 in the translation repression of its targets. This defect in translation is also accompanied by an increase in the global abundance of m^6^A-modified mRNAs, confirming the intimate link existing between the number of m^6^A sites and the instability of the targeted mRNA [26]. This activity of YTHDF2 in mRNA destabilization requires both its N- and C-terminal regions as over-expression of full-length YTHDF2 leads to decay of m^6^A containing mRNAs, while expression of only the N-terminal or the C-terminal region does not have the same effect [26]. This role of YTHDF2 in the degradation of m^6^A-containing mRNAs is further supported by its localization in Processing bodies (P-bodies) in which YTHDF2 co-localizes with DCP1a and DDX6 proteins known to be involved in mRNA decapping [26]. YTHDF2 also directly interacts with CNOT1, the scaffolding subunit of the CCR4-NOT mRNA deadenylase [43]. This interaction relies on the SH domain from CNOT1 and the YTHDF2 N-terminal domain, which is also responsible for the localization of at least YTHDF2 to P-bodies (Figure 3; [43]). Interestingly, Pho92, the only YTH-containing protein from *S. cerevisiae*, also interacts with Pop2, another component of the CCR4-NOT complex [68]. The similarities between human YTHDF2 and *S. cerevisiae* Pho92 are emphasized by (1) the ability of human YTHDF2 gene (but not YTHDC1) to complement for the deletion of PHO92 gene in *S. cerevisiae* and (2) the role of Pho92 as an enhancer of mRNA decay [68]. Altogether, this suggests that the main role of YTHDF2 is in the regulation of m^6^A-containing mRNA decay and that this role has been conserved throughout evolution. Interestingly, upon heat shock, YTHDF2 relocalizes to the nucleus and this is accompanied by a specific increase of m^6^A in the 5′ UTR of stress-inducible mRNAs and an increased ribosome occupancy in their coding region [69]. This could then contribute to the stimulation of translation by the direct recruitment of the translation initiation factor 3 complex (eIF3) to m^6^A sites located within mRNA 5′ UTRs [70].

Despite the strong sequence similarity with YTHDF2, YTHDF1 knock-down does not affect the m^6^A/A ratio, indicating that this protein is unlikely to be involved in m^6^A-containing mRNA decay [42]. On the contrary, YTHDF1 seems to enhance the translation efficiency of a population of transcripts encoded by around 1200 genes, to which it associates in an m^6^A-dependent manner. This mechanism is likely to occur through the recognition of m^6^A sites by YTHDF1 within the 3′ UTR of mRNAs on the one hand, as well as with the 40S subunit and components of the eIF3 complex bound in the vicinity of the start codon on the other (Figure 3; [42]). This mechanism differs from the one described above for 5′ UTR m^6^As that directly recruit eIF3 under stress conditions and that is independent of YTHDF1 [70]. Hence, the effect of YTHDF1 on translation of m^6^A-mRNAs may be limited to a small subset of RNAs depending on various physiological situations. In the nervous system for example, transcriptome-wide mapping of YTHDF1-binding sites, combined with nascent protein labelling, revealed that YTHDF1 enhances translation of key hippocampal m^6^A-methylated mRNAs in response to neuronal stimulation, thus contributing to learning and memory [71].

Several observations indicate that YTHDF3 interacts with both YTHDF1 and YTHDF2 proteins and thereby works together with those factors to up-regulate translation (YTHDF1) or enhance degradation of mRNAs (YTHDF2), respectively [46,72]. Indeed, PAR-CLIP coupled to RIP-seq showed that YTHDF3 shares more than half of its targets with YTHDF1 but also YTHDF2, which is not surprising considering the high degree of sequence identity (86% to 89%) between the YTH domains from these three proteins [46,72]. However, human YTHDF3 might be also mobilized independently of YTHDF1 and YTHDF2 under specific conditions. Notably YTHDF3 co-localizes exclusively with the stress granules under oxidative stress, whereas YTHDF1 and YTHDF2 retain their cytoplasmic localization with only marginal presence in stress granules. In this specific context, YTHDF3 selectively recognizes a pool of oxidative stress-induced methylated mRNAs in order to mediate triaging of mRNAs from the translatable pool to stress granules (Figure 3; [73]). Overall, this indicates that all three YTHDF family proteins may participate in a complex regulatory mechanism that results first in a higher translational efficiency of m^6^A-mRNAs followed by their rapid degradation. This complex interplay between these three YTHDF proteins and interacting proteins involved in these different cellular processes will need to be clarified in the future.

Interestingly, m^6^A marks are not restricted to cellular RNAs but are also found in viral mRNAs, where they are recognized by the YTHDF1–3 proteins [74,75,76,77,78]. Several recent studies have focused on the roles of m^6^A and YTHDFs on the regulation of viral infection leading to the description of various mechanisms. For instance, the recognition of m^6^A marks on viral mRNAs by YTHDFs has been shown to block reverse transcription of Zika virus genome [77]. In the case of Hepatitis C virus, YTHDFs inhibit HCV infection without affecting RNA replication by a mechanism that could be common to most Flaviviridae (Figure 3; [75]). Finally, the role of YTHDFs on HIV-1 infection is not clear, as Tirumuru et al. have shown that YTHDFs inhibit infection by decreasing reverse transcription of the viral genome [78] while Kennedy et al. presented data supporting a role of YTHDFs as enhancers of the expression of both viral RNA and proteins and of viral replication [76]. Further studies are then clearly needed to clarify the role of m^6^A marks on the infection of human cells by various viruses.

## 4. *Schizosaccharomyces pombe* Mmi1, a Fission within YTH Family Proteins

Comparative genomics led to the conclusion that the m^6^A ‘writers’ homologous to human METTL3 and METTL14 proteins are absent in *S. pombe* fission yeast [79,80]. However, *S. pombe* Mmi1 (for Meiotic mRNA interception 1), a YTH-domain containing protein, is present in this organism. Mmi1 has the same organization as YTHDFs with a low-complexity N-terminal region rich in Prolines and charged residues followed by a C-terminal YTH (Figure 2).

Mmi1 is an RNA binding protein that specifically recognizes a consensus U(U/C/G)AAAC motif, present in multiple copies within larger regions called Determinant of Selective Removal (DSR). This motif radically differs from the m^6^A consensus binding site recognized by canonical YTH domains [81]. Mmi1 YTH domain is responsible for the binding to the DSR and is assisted in this function by a low complexity region located upstream this YTH domain [82]. Although this domain is structurally similar to YTH domains from other proteins such as YTHDC1, YTHDF2, and Pho92, it is using a different surface to interact with its RNA consensus motif (Figure 4A). This binding surface is located on the opposite face of the Mmi1 YTH domain compared to the region involved in m^6^A binding by classical YTH domains (Figure 4A; [82,83,84]). This region is conserved in fission yeasts (*S. pombe*, *Schizosaccharomyces japonicus*, *Schizosaccharomyces octosporus*, and *Schizosaccharomyces cryophilus*) but not in other YTH containing proteins. Mmi1 YTH domain is incapable of binding a GGm^6^AC containing RNA, and N6-methylation at any position of the DSR motif weakens binding to the RNA [83]. Similarly, the YTH domains from YTHDC1, YTHDF2 and Pho92 do not bind DSR consensus motif. Detailed comparison of the Mmi1 pocket corresponding to m^6^A binding site in other YTH domain proteins shows that two Trp residues are conserved while the position corresponding to Trp, Leu, or Tyr in canonical YTH domains is occupied by a His in Mmi1. However, two main differences that could explain the Mmi1 YTH domain inability to bind m^6^A containing RNAs have been observed. First, the amino acid corresponding to the Asn, Asp, or His residues forming a hydrogen bond with N1 atom of the adenosine ring is substituted by Ala in Mmi1. Second, while the surface surrounding the m^6^A binding pocket is positively charged in canonical YTH domains, the corresponding region in Mmi1 is negatively charged, which is not favorable for RNA binding [83]. Hence, Mmi1 is an exception among YTH domains as it cannot recognize m^6^A modification and interacts with RNA in a completely different manner.

Mmi1 is a key cell fate regulator that promotes degradation of meiosis-specific transcripts in mitotic cells and hence seems to fulfill a similar function as human YTHDF2 and *S. cerevisiae* Pho92, namely the regulation of the stability of its RNA targets [26,68,81,85]. During mitosis, Mmi1 localizes in the nucleus and stably interacts with Erh1, a small protein with still unclear function, to form the Erh1-Mmi1 Complex (EMC) (Figure 4B; [86]). The EMC associates with MTREC (Mtl1-Red1 core), a multi-subunit complex containing, among others, the zinc-finger protein Red1, the Mtr4-like RNA helicase Mtl1, and the serine and proline-rich protein Pir1/Iss10 [86,87,88,89,90]. EMC and MTREC associate and cooperate with the Rrp6 subunit of the nuclear exosome for the selective elimination of meiotic DSR-containing transcripts (Figure 4B; [81,91,92]). In addition, EMC sequesters meiotic mRNAs in nuclear foci, preventing their export to the cytoplasm and their translation [90]. Therefore, Mmi1 silences the expression of meiotic genes in at least two different ways, through RNA degradation and nuclear retention.

Recent works have also reported a tight association between Mmi1 and the CCR4-NOT complex, resulting in its recruitment to meiosis specific transcripts *in vivo* as well as in the stimulation of deadenylation activity in vitro [44,86,93,94]. However, this function of the CCR4-NOT complex is not mandatory for the degradation of meiotic mRNAs. Rather, Mmi1 recruits CCR4-NOT complex to promote ubiquitination and down-regulation of its own inhibitor, the meiosis inducer Mei2, via the Not4/Mot2 E3 ubiquitin ligase subunit [89]. This regulatory circuit preserves the activity of Mmi1, ensuring efficient meiotic mRNA degradation in mitotic cells. Upon entry into meiosis, instead, Mmi1 is sequestered in an RNP (ribonucleoprotein) complex formed by the Mei2 protein and the long noncoding meiRNA, thereby allowing expression of the meiotic program (Figure 4B; [81]). Mei2 may also inactivate Mmi1 at the mRNA level as it binds to its transcripts during early meiosis [95]. Further studies are needed to clarify the relationships between these two main effectors of the mitosis-meiosis switch in *S. pombe*.

## 5. Conclusions

The technological progresses made during the last decade have allowed the identification of RNA modifications present at low abundance and on RNA species with limited stability such as mRNAs and long noncoding RNAs in eukaryotic cells. This research has created the basis for the characterization of proteins and multi-protein complexes involved in the deposition, elimination, and detection of these modifications. So far, much of the studies have focused on the most abundant modification, namely m^6^A. As a consequence, the best characterized m^6^A ‘readers’ are YTH-containing proteins and studies in human cells have shown that the five human YTH proteins are affecting mRNA fates such as splicing, export, translation, and decay by recruiting various protein partners. However, there is increasing evidence that proteins lacking YTH domains can also selectively recognize m^6^A marks. This is indeed the case for instance of the eIF3 initiation factor [70], the FMR1 RNA binding protein, which loss results in fragile X-linked mental retardation syndrome [28,96] or the insulin-like growth factor 3 mRNA-binding proteins IGF2BPs [97]. Undoubtedly, epitranscriptomics is a highly dynamic field and future work will identify new ‘readers’ of epitranscriptomics marks as has recently been the case for ALYREF, an m5C ‘reader’ [98], paving the way for the description of highly sophisticated mechanisms to control gene expression at the mRNA level.

## Figures and Tables

**Figure 1 genes-10-00049-f001:**
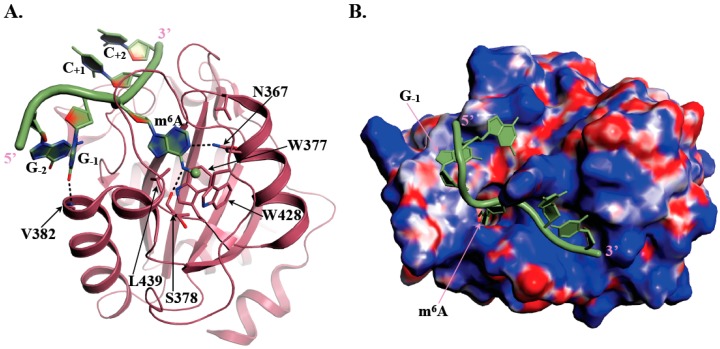
Recognition of m^6^A (N6-methyladenosine) containing RNAs by human YTHDC1 YTH domain. (**A**) Ribbon representation of a GGm^6^ACC RNA oligonucleotide (green) bound to human YTHDC1 (PDB code: 4R3I; [37]). The methyl group grafted on N6-adenosine is shown as a sphere. The side chains from residues involved in the formation of the m^6^A aromatic cage and the hydrogen bonds responsible for specificity of m^6^A as well as for increased affinity of YTHDC1 for RNAs harboring a G just upstream of the m^6^A mark are shown as sticks. Hydrogen bonds are depicted by black dashed lines. (**B**) Mapping of the electrostatic surface at the surface of human YTHDC1 YTH domain with the GGm^6^ACC RNA oligonucleotide shown in green. Positively (8 kBT/e) and negatively (−8 kBT/e) charged regions are colored blue and red, respectively. The potential was calculated using the CHARMM-GUI server [39].

**Figure 2 genes-10-00049-f002:**
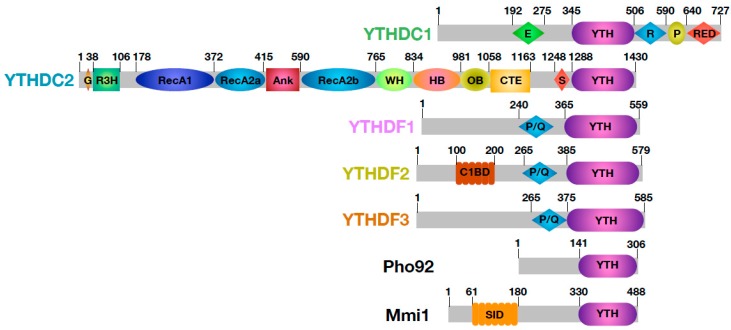
Schematic representation and domain composition of human, *Saccharomyces cerevisiae* (Pho92) and *Schizosaccharomyces pombe* (Mmi1) YTH-domain containing proteins. The predicted or experimentally determined limits of domains are indicated. E: Glu-rich domain. R: Arg-rich domain. P: Pro-rich domain. RED: Arg/Glu/Asp-rich domain. G: Gly-rich domain. R3H: small domain containing an invariant Arg sported from a highly conserved His by three residues. RecA1 and RecA2: RecA domains found in helicases. Ank: Ankyrin repeats. WH: Winged-helix domain. HB: Helical bundle. OB: OB-fold. CTE: C-terminal extension. S: Ser-rich domain. P/Q: Pro and Gln-rich domain. C1BD: CNOT1 binding domain. SID: Self-Interacting domain.

**Figure 3 genes-10-00049-f003:**
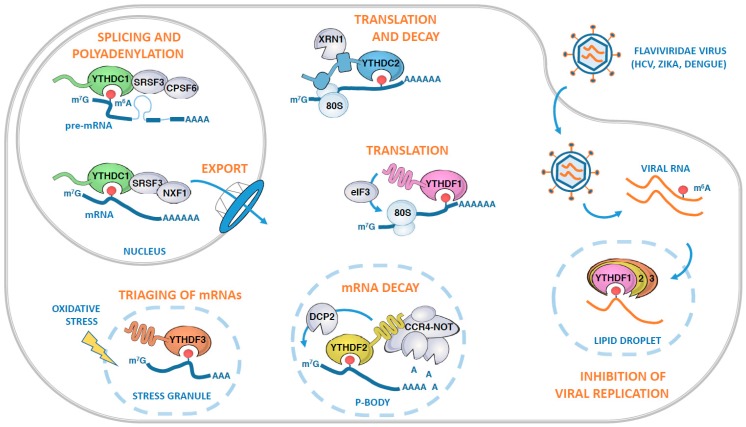
Roles of human YTH-containing proteins in various aspects of messenger RNA (mRNA) fates. In the nucleus, YTHDC1 recognizes m^6^A-modified pre-mRNAs and orchestrates their splicing, polyadenylation, and nuclear export through its association with SRSF3, CPSF6, and NXF1. Once in the cytosol, the modified mRNAs can be bound by YTHDC2, which in turn recruits both the ribosome and the XRN1 exoribonuclease. Alternatively, the mRNA can be targeted by the YTHDF proteins, either to be actively translated in an YTHDF1-dependent manner, or subjected to mRNA decay through YTHDF2 and its ability to recruit the CCR4-NOT deadenylase complex. YTHDF2-mediated mRNA decay is likely to occur in Processing bodies (P-bodies) where it co-localizes with the decapping enzyme DCP2. Following oxidative stress, m^6^A-modified mRNAs can also be recognized by YTHDF3, which facilitates the triaging of mRNAs into the stress granules. The YTHDF proteins can also target viral m^6^A-modified RNAs during infection, as illustrated by the case of hepatitis C virus (HCV) infection. During infection, the YTHDF proteins relocalize to lipid droplets, sites of viral assembly, and sequester m^6^A-modified HCV RNAs, preventing their interaction with HCV core protein and subsequent virus particle production.

**Figure 4 genes-10-00049-f004:**
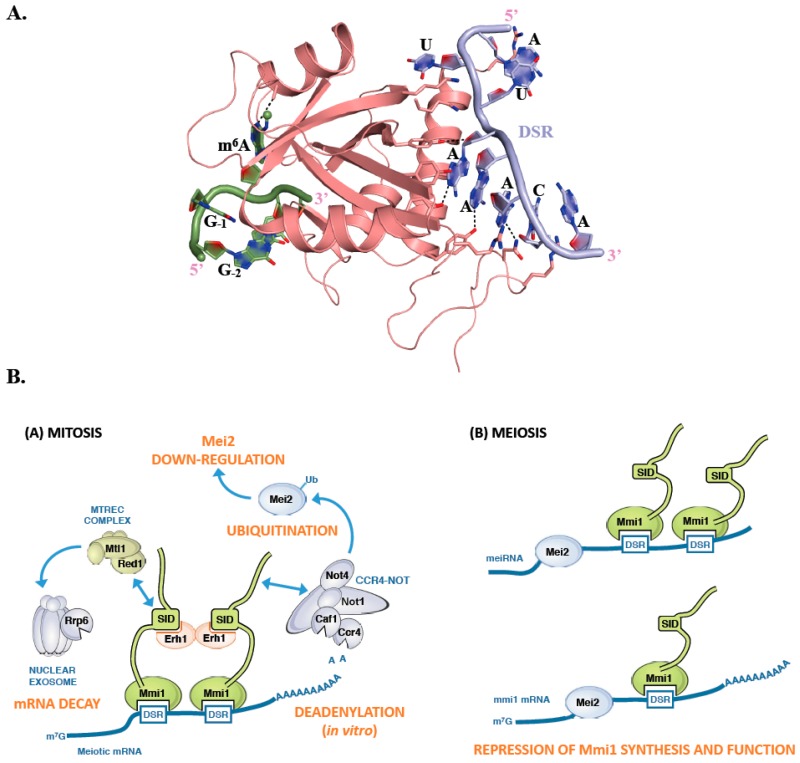
*S. pombe* Mmi1 contains an atypical YTH domain. (**A**) Ribbon representation of the complex between *S. pombe* Mmi1 YTH domain (pink) and a Determinant of Selective Removal (DSR) RNA sequence (light blue; [84]). An m^6^A containing RNA fragment (green) has been modeled by superimposing the crystal structure of RNA-bound YTH domain from YTHDC1 onto the structure of *S. pombe* Mmi1. Some residues and hydrogen bonds important for the interaction between *S. pombe* Mmi1 and the RNA DSR sequence are shown as sticks or black dashed lines, respectively. (**B**) Cartoons summarizing how Mmi1 controls the decay of meiotic mRNAs during mitosis or is inactivated during meiosis.

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
