# Peer review of "m6A mRNA Destiny: Chained to the rhYTHm by the YTH-Containing Proteins"

_genes, 2019, doi:10.3390/genes10010049_

Reviewer 1 Report

It is an interesting manuscript but needs some corrections: grammar should be re-checked and some other small corrections should be introduced to improve the manuscript:

1.     Line 36: double space between “regulation  new”

2.     Line 37:  “Indeed, similar” instead of “Indeed, similarly”

3.     Lines 40-41: “does not alter base pairing” - reference missing

4.     Line 69: No citation supporting the sentence, for example a review paper by Bhat et al. published in 2018 in Genes.

5.     Line 71: YTHDF1/3 should be replaced by YTHDF 1-3

6.     Line 84 and 124: Two different PDB codes, it should be unified or described why in Figure 1  the PDB code is different from this used in the text. Citation is also missing.

7.     Lines 89, 98, 109: The description of Figure 1X is not clear.

8.     Line 100: The sentence: “A large majority…’’ should be moved to the introduction chapter as the motif is already mentioned there.

9.     Line 176: Text concerns Figure 3 instead of Figure 2. Please change accordingly.

10.  Line 207: In the sentence: “Inactivation of either …” a term ‘enzyme’ should be replaced with ‘complex’.

11.  Line 334: “consensus motif U(U/C/G)AAAC motif”

12.  Figure 2: In the schematic representation of YTH proteins S. cerevisiae and plants YTH-domain containing proteins should be included.

13.  Although the authors mentioned the presence of 13 members of the YTH family in Arabidopsis thaliana, these proteins are completely excluded from any further discussion concerning the YTH protein structure and function. This should be corrected.

14.  Line 329: The authors claim that m6A writers are absent in S. pombe, but this statement seems to be too strong and needs an addition explanation. Bujnicki et al. (as cited by the authors) report the presence of MTases from the lineage C in S. pombe and they show that members of this lineage have all conserved motifs of MTAses and may act as active MTAses although it is not experimentally proven.

15. Chapter 4 is focused on the Mmi1 protein which as the authors write is unable to bind m6A  (“inability to bind m6A containing RNAs” (Lines 349-350)). It would be better focused more on comparison between S. cerevisiae (Pho92) and S. pombe

Author Response

Thank you very much for this supportive assessment of our work and for your suggestions to improve our manuscript. We re-checked grammar and corrected minor points raised. Please find below a point-by-point response to your comments/suggestions.

1.     Line 36: double space between “regulation  new”

This has been done.

2.     Line 37:  “Indeed, similar” instead of “Indeed, similarly”

This has been done.

3.     Lines 40-41: “does not alter base pairing” - reference missing

A reference from Roost and coworkers has been added. This article describes the NMR structures of an unmodified GGACU RNA duplex and of its m6A modified version, which are very similar. The authors conclude in the first sentence of the discussion that « the m6A-U pair in a stable RNA duplex is paired in canonical configuration, with two Watson-Crick hydrogen bonds. »

4.     Line 69: No citation supporting the sentence, for example a review paper by Bhat et al. published in 2018 in Genes.

The reference for the original description of YTH domain by Stoilov has been added as well as the reference by Bhat and coworkers in Genes (2018).

5.     Line 71: YTHDF1/3 should be replaced by YTHDF 1-3

This has been done both for YTHDC 1-2 and YTHDF 1-3.

6.     Line 84 and 124: Two different PDB codes, it should be unified or described why in Figure 1  the PDB code is different from this used in the text. Citation is also missing.

We cite 2 PDB codes in the review. The first one (PDB code 2YUD, line 81) corresponds to the structure of the YTHDC1 YTH domain solved by NMR in the absence of ligand. This is the first structure of this domain that has ever been solved. Unfortunately, the paper describing this structure has never been published but the coordinates have been deposited and are available. 

The second PDB code cited (PDB code: 4R3I, line 125) corresponds to the structure of the YTHDC1 YTH domain bound to an RNA oligonucleotide. The paper describing this structure is cited many times in the review.

7.     Lines 89, 98, 109: The description of Figure 1X is not clear.

This has been clarified.

8.     Line 100: The sentence: “A large majority…’’ should be moved to the introduction chapter as the motif is already mentioned there.

As this statement is linked to the next sentence indicating that YTHDC1 YTH domain has a higher affinity for Gm6AC motif compared to Am6AC, we think that moving this sentence to the introduction is not appropriate.

9.     Line 176: Text concerns Figure 3 instead of Figure 2. Please change accordingly.

This has been done.

10.  Line 207: In the sentence: “Inactivation of either …” a term ‘enzyme’ should be replaced with ‘complex’.

We have changed enzyme by holoenzyme, which is definitely more appropriate than enzyme.

11.  Line 334: “consensus motif U(U/C/G)AAAC motif”

This has been done.

12.  Figure 2: In the schematic representation of YTH proteins S. cerevisiae and plants YTH-domain containing proteins should be included.

Thanks for noticing this omission. We have added a schematic representation of S. cerevisiae Pho92 protein to Figure 2. As there are 13 YTH proteins in A. thaliana, we believe that adding their schematic representation would dilute the message that we want to give in Figure 2. Furthermore, as mentioned in your next comment (13), we do not mention the plant YTH proteins as the review contributed by Bhat and colleagues to the same GENES special issue deals with m6A machineries in plant and hence already addresses this point. 

13.  Although the authors mentioned the presence of 13 members of the YTH family in Arabidopsis thaliana, these proteins are completely excluded from any further discussion concerning the YTH protein structure and function. This should be corrected.

We agree that we did not discuss the plant YTH proteins in this review. This was a deliberate choice as Bhat and colleagues contribute an excellent review on m6A machineries in A. thaliana in which they deal with YTH domain proteins as m6A readers. To clarify this point, we have added the following sentence in the second paragraph of the « YTH, a building block governing the fates of m6A containing mRNAs » section:

« As mentioned above, bioinformatics analyses have identified 13 YTH-containing proteins in A. thaliana, which roles are being clarified. As this topics has been nicely discussed by Bhat et al in this special issue, these plant YTH proteins will not be discussed in this section. »

14.  Line 329: The authors claim that m6A writers are absent in S. pombe, but this statement seems to be too strong and needs an addition explanation. Bujnicki et al. (as cited by the authors) report the presence of MTases from the lineage C in S. pombe and they show that members of this lineage have all conserved motifs of MTAses and may act as active MTAses although it is not experimentally proven.

This is correct that in this article by Bujnicki and coworkers, they mention the identification of an S. pombe protein with similarity to a human protein from lineage C. This human protein is now known as METTL4 and it function is still elusive. It is then currently well accepted that S. pombe lacks homologues of human METTL3 (lineage A in Bujnicki and coworkers paper) and METTL14 (lineage B in Bujnicki and coworkers paper) proteins (see review by Fu et al; Nature Reviews Genetics, 2014) and so far, to our knowledge, no m6A has been found in S. pombe mRNAs. To clarify this point, we have updated the text as follow: « Comparative genomics led to the conclusion that the m6A writers homologous to human METTL3 and METTL14 proteins are absent in S. pombe fission yeast »

15. Chapter 4 is focused on the Mmi1 protein which as the authors write is unable to bind m6A  (“inability to bind m6A containing RNAs” (Lines 349-350)). It would be better focused more on comparison between S. cerevisiae (Pho92) and S. pombe

The inability of YTH domain from Mmi1 to interact with m6A-containing RNA oligonucleotide has been discussed in the original papers describing crystal structure of Mmi1 YTH domain bound to RNA (Wang et al; Nucleic Acids Research; 2016 / Wu et al; BBRC; 2017 / and in a recent review (Liao et al; Genomics Proteomics Bioinformatics; 2018) .We then preferred discussing the functional role of this unusual YTH-containing proteins than comparing it to canonical YTH proteins such as Pho92.

Reviewer 2 Report

m6a modification is the most important and widely prevalent RNA modification which is involved in many essential biological pathways. For now, the functional and structural studies is very hot and I am also very interested in this field. This review focus on the m6a reader and summarizes structures and functional progress. The manuscript elaborate clear and comprehensively. But I still have some questions.

1. I do not quite understand the meaning of the title. Can you explain what is "rhYTHm"?

2. For the yeast m6a writer, I know that saccharomyces cerevisiae has ime4-mum2-slz1 complex. I do not think m6a writer is absent in Schizosaccharomyces pombe. If it is absent, how can the m6a methylation happen?

Author Response

We thank you for your positive evaluation of our manuscript. Please find below a point-by-point response to your questions.

I do not quite understand the meaning of the title. Can you explain what is « rhYTHm"?

We choose this title as rhythm contains the contiguous YTH letters and because m6A recognition by YTH domains affects the beat (or rhythm) of the mRNA fate. Indeed, it is now clear that an mRNA containing an m6A modification will be degraded faster than the same unmodified mRNA, as well as it will exhibit an accelerated translational rate.

2. For the yeast m6a writer, I know that saccharomyces cerevisiae has Ime4-Mum2-Slz1 complex. I do not think m6A writer is absent in Schizosaccharomyces pombe. If it is absent, how can the m6a methylation happen?

It is then currently well accepted that S. pombe lacks homologues of human METTL3 and METTL14 proteins (see review by Fu et al; Nature Reviews Genetics, 2014) and so far, to our knowledge, no m6A has been found in S. pombe mRNAs. This is consistent with the fact that S. pombe YTH protein (Mmi1) does not recognize m6A containing mRNAs. To clarify this point, we have updated the text as follow: « Comparative genomics led to the conclusion that the m6A writers homologous to human METTL3 and METTL14 proteins are absent in S. pombe fission yeast »

Reviewer 3 Report

This is an excellent review on YTH domain-containing proteins as m6A readers. The manuscript is well documented and informative. 

I have only minor comments below:

- Figure 1A: 

L438 should be L439.

- Figure number in the text should be corrected.

Page 2, Line 89: Figure 1X should be 1A.

Page 3, Line 98: Figure 1X should be 1A.

Page 3, Line 109: Figure 1X should be 1A.

Page 5, Line 176: Figure 2 should be 3.

Page 8, Line 340: Figure 4 should be 4A.

Author Response

We thank you for your positive evaluation of our manuscript and your suggestions to improve its overall quality. All your comments have been addressed in the revised version.